

# Reproductive ecology of the endangered Beal's-eyed turtle, *Sacalia bealei*

Liu Lin[1], Qingru Hu[1], Jonathan J. Fong[2], Jiangbo Yang[1], Zhongdong Chen[3], Feiyu Zhou[3], Jichao Wang[1], Fanrong Xiao[1] and Haitao Shi[1]

[1] Ministry of Education Key Laboratory for Ecology of Tropical Islands, College of Life Sciences, Hainan Normal University, Haikou, Hainan, China
[2] Science Unit, Lingnan University, Hongkong, China
[3] Administration Bureau, Fujian Huboliao National Nature Reserve, Zhangzhou, Fujian, China

## ABSTRACT

The Beal's-eyed turtle (*Sacalia bealei*) is endemic to southeastern China and endangered due to poaching and habitat loss. Knowledge of *S. bealei* ecology is lacking and this study provides baseline information of its reproduction in a natural environment. We studied the reproductive ecology of *S. bealei* using X-ray, spool-and-line tracking, and direct observation. Six nesting females were successfully tracked and their nesting behaviors are documented in detail. Females produced a mean clutch size of 2.2 eggs (range 1–3). The hard-shelled eggs were ellipsoidal with a mean length of 45.50 mm, a mean width of 23.20 mm, and mean weight of 14.8 g. The relative clutch mass was 9.47%, while the relative egg mass was 4.60%. The mean incubation period was 94.7 days with a mean nest temperature of 25.08 °C. Hatchlings had a mean weight of 9.7 g, carapace length of 40.1 mm, carapace width of 33.3 mm, carapace height of 17.4 mm, plastron length of 31.6 mm, and plastron width of 25.4 mm. The results of this study provide important information to inform conservation plans and ex-situ breeding for this endangered species.

## INTRODUCTION

The reproductive biology of a species is an important component of its overall life history strategy (*Gibbons, 1982*). Thus, understanding reproductive ecology is important for turtle management and conservation (*Tucker & Moll, 1997*; *Horne et al., 2003*). Studies of turtle reproductive ecology have revealed important direct and indirect effects on fitness and demography (*Bobyn & Brooks, 1994*; *Weisrock & Janzen, 1999*; *Valenzuela, 2001*; *Spencer & Thompson, 2003*; *Janzen, Tucker & Paukstis, 2007*).

The Beal's-eyed Turtle (*Sacalia bealei*), endemic to southeastern China, is distributed in Guangxi, Guangdong, Fujian, Anhui, Guizhou, Jiangxi Provinces, and Hong Kong (*Shi et al., 2008*). It is listed as endangered on the IUCN Red List (*van Dijk et al., 2012*) and Appendix II of CITES. Poaching pressure on *S. bealei* is strong, with the pet trade price increasing from 1500 RMB/kg in 2014 to 4200 RMB/kg in 2015 (*Hu, 2016*). Due to illegal poaching and trade, *S. bealei* has become extremely rare in the field

Corresponding author
Haitao Shi, haitao-shi@263.net

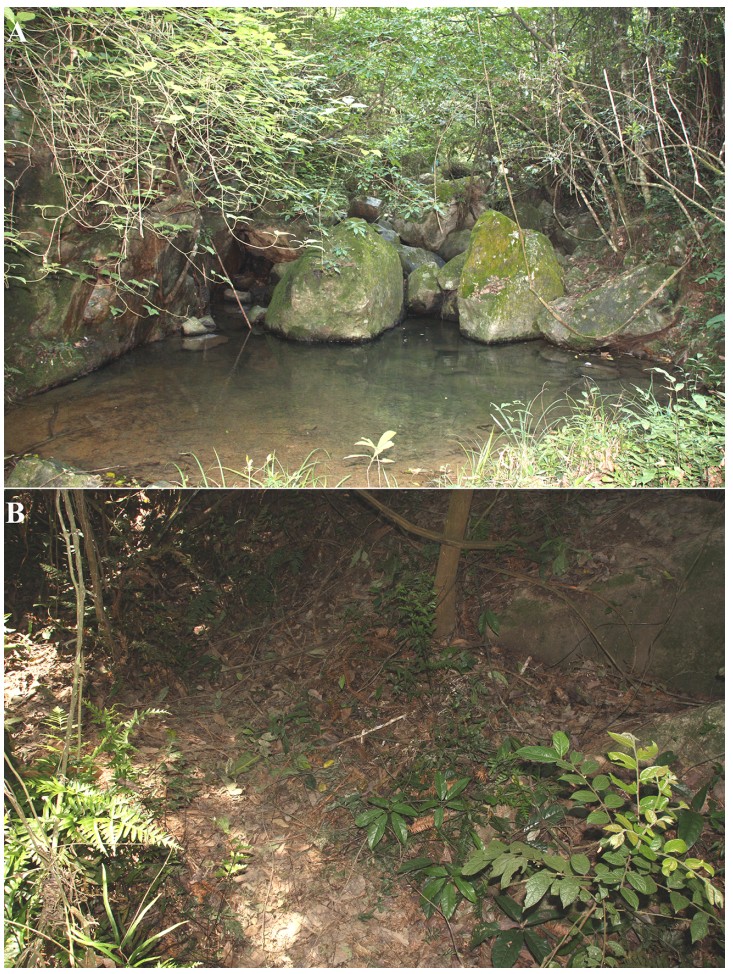

**Figure 1 Natural habitat of *S. bealei*.** (A) An ideal stream habitat with many big stones; (B) A typical nesting site in the nearby forest, covered by heavy canopy. Photos credit: Liu Lin.

(*Shi, O'connell & Parham, 2005*; *Gong et al., 2017a*). Life history of this species is poorly understood, with only a few observations on diet and reproduction in captivity (*Zhang, Zong & Ma, 1998*; *Gong et al., 2017b*). This study provides baseline information on the reproductive biology of *S. bealei* in a natural environment.

## METHODS

### Study site

We conducted field research in Huboliao National Nature Reserve, Fujian Province, China (117°12′42″–117°22′45″E; 24°30′05″–24°56′2″N). The mean annual temperature in Huboliao is 21.1 °C, with the lowest temperature in January (mean 10.9 °C) and the highest in July (mean 26.7 °C). The mean rainfall is 1733.5 mm, with a mean relative humidity of 81.4%. The major vegetation types in this reserve are evergreen broad-leaved forest, mixed forest, and bamboo forest (*Fan, 2001*) (Fig. 1).
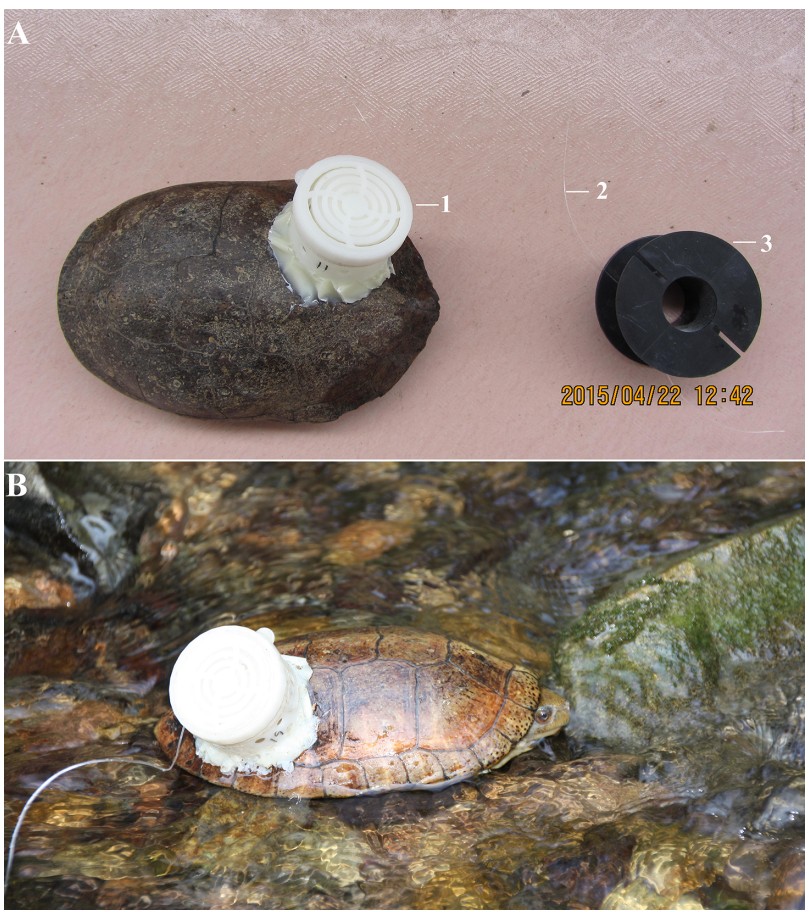

**Figure 2 Tracked female turtles with a spool-and-line tracker.** (A) The components of the tracker. 1, The white box with an internal line spool; 2, the fishing line; 3, the black external line spool for retrieving the line; (B) a female turtle in the wild with the tracker on her carapace. Photo credit: Qingru Hu (A), Liu Lin (B).

## Methods

In late March 2015, when turtles had completed their hibernation (from early November to mid March), we used traps to capture them. When females were captured, we brought them back to the field station for further inspection. We used portable X-ray radiography (BJI—UJ) to confirm the presence and number of oviductal eggs. If shell-eggs were found, a plastic spool-and-line tracker was attached to the carapace of the female (Fig. 2). The spool-and-line tracker was composed of two parts: (1) a white box (diameter 35 mm, height 23 mm) with an internal line spool (diameter 30 mm, height 19 mm), fastened to the carapace using epoxy resin, (2) a black external line spool (diameter 39 mm, height 23 mm), which was attached to the line and used afterwards for retrieval. The tracker weighed 8 g and contained 70 m fishing line. We designed the tracker and have successfully used it to track daily movement of the red-eared slider (*Trachemys scripta elegans*) during the nesting period (*Li, 2013*; *Yang, 2014*). In total, 10 females with evidence of shell-eggs were tracked. We took morphological measurements of these individuals (weight, carapace length, carapace width,

body height, plastron length, and plastron width) following methods in *Xiao, Shi & Sun (2014)*.

From 8:00 am each day, we followed the fishing line of the 10 females to track their movement. The fishing line was retrieved carefully and spooled back into the white box every day to ensure the line was not used up and so the turtles could move freely. The presence of fishing line on land indicated that the individual was attempting to nest. When we encountered such a situation, we took notes on the nesting and egg-laying behavioral patterns, minimizing disturbance by observing using binoculars from 8–10 m away. After the turtle laid eggs and returned back to the stream, we checked the nest, recording (1) nest chamber characters (e.g., size and nest materials), (2) clutch size and egg measurements (length, width, and weight), (2) distance crawled by the female before nesting (following the fishing line), and (4) straight distance between the nest and the stream bank. Developing and undeveloping eggs were distinguished by the presence of a white spot on the eggshell. Relative egg mass (REM) (mean egg weight/body weight before oviposition) and relative clutch mass (RCM) (clutch weight/body weight before oviposition = REM × clutch size) were calculated for each gravid female as estimates of reproductive effort (*Wang et al., 2011*). To monitor the temperature and humidity of the nest during incubation, we buried a data logger (HOBO U23-001, Onset Computer Inc., Bourne, MA, USA) at 10 cm distance to the nest and at the same depth. To prevent predation, we placed a wire net above the nest.

We checked the nests at least twice a week during incubation. When hatchling turtles emerged from the nests, we measured their weight, carapace length, carapace width, plastron length, and body height. Hatchlings were taken back to the field station and released back to the stream when the yolk sac was completely absorbed. Data loggers were retrieved after hatchling emergence, and the data were downloaded by HOBOware Graphing & Analysis Software (Onset Computer Inc., Bourne, MA, USA).

The field study was approved by Administration and Services Center of Nanjing County, Fujian Province (No. 20141203NJ0173). The animal ethics was approved by Animal Research Ethics Committee of Hainan Provincial Education Center for Ecology and Environment, Hainan Normal University (No. HNECEE-2014-003).

### Data analysis

Statistical analyzes were performed in SPSS 19.0. Descriptive statistics were expressed by mean ± standard error. Significance level was set to 0.05. We used linear regression to investigate the relationship between female body size and clutch or egg size.

## RESULTS

### Timing and behavior sequence of nesting

The 10 gravid females captured in this study had a mean weight of 329.6 ± 11.8 g, carapace length of 136.3 ± 1.5 mm, carapace width of 94.4 ± 1.1 mm, carapace height of 51.3 ± 0.8 mm, plastron length of 121.7 ± 1.5 mm, and plastron width of 78.8 ± 0.9 mm ($n = 10$). Among them, six individuals were successfully tracked. The remaining four individuals were lost because the fishing line broke. The nesting activities lasted from May 3–31.

For the six females, we recorded 10 nesting activities—four failed attempts and six successful nesting. All observed nesting activities happened on rainy days and egg laying only happened at night. The behavior sequences of nesting could be categorized into five successive steps: (1) nest-site selection, (2) chamber excavation, (3) egg laying, (4) nest covering, and (5) returning to water.

During the nest-site selection, females emerged from water and made their way into the forest. Movement was not continuous, but instead occurred in stops and starts. During the pauses, individuals often elevated their head a few centimeters above the carapace and move side-to-side a few times, possibly as a vigilance behavior. While moving, the ground substrate did not impede their movement—they crawled along rocks, fallen wood, thick leaves, and gentle slopes. Often, females would attempt to dig nests one to two times before they chose the final site.

After finding the right site, females excavated the nest chamber, alternating between the two hindlimbs. After completing excavation, females used the rear portion of its carapace to hit the nest four to six times. The chamber excavation lasted 40–60 min.

After resting approximately 20 min, females would start laying eggs with hindlimbs straddling the chamber opening. To expel eggs, forelimbs and head would completely extend then simultaneously withdraw. Body tremors were obvious when the eggs were released into the nest. In total, egg laying lasted 8–15 min. No vigilance behaviors were observed during this sequence.

Nest covering began immediately after laying the last egg. Soil and leaves were packed down using alternate, backward movements of the hindlimbs, followed by a bobbing motion of the plastron that further compressed the soil and leaves into the nest chamber. Nest covering lasted 40–70 min.

After covering the nest, females immediately returned to water. While returning to the water, similar behaviors to those when choosing a nest-site were exhibited, such as movement in stops and starts, and crawling along rocks and fallen wood.

## Nest and egg characteristics

Each of the six tracked females laid eggs in a nest. The number of eggs in each nest was consistent with the number of oviductal eggs seen in the X-ray, showing all eggs were laid at once. Straight distance from nest to the stream bank was $8.55 \pm 1.23$ m ($n = 6$), while straight distance from nest to the sites where they emerged from and returned to water, was $11.47 \pm 1.31$ m and $16.55 \pm 2.93$ m ($n = 6$), respectively. Females crawled a distance of $42.67 \pm 7.18$ m ($n = 6$) to choose a nesting site.

Nests were well camouflaged by leaves and soil (Fig. 3), making it difficult to find nests without the help of fishing line. The shape of nests was hemispherical, with a diameter of 70–100 mm and depth of 55–70 mm. Due to the relatively shallow depth, eggs were often half-buried by soil then covered by leaves.

The mean clutch size was $2.2 \pm 0.3$ (range 1–3, $n = 6$). The hard-shell eggs were ellipsoidal, and weighed $14.8 \pm 0.8$ g, with a length of $45.5 \pm 1.0$ mm and width of $23.2 \pm 0.5$ mm ($n = 13$). A negative, linear correlation was found between mean egg length and female plastron width (Pearson correlation coefficient = 0.934, $R^2 = 0.934$, $P < 0.05$),

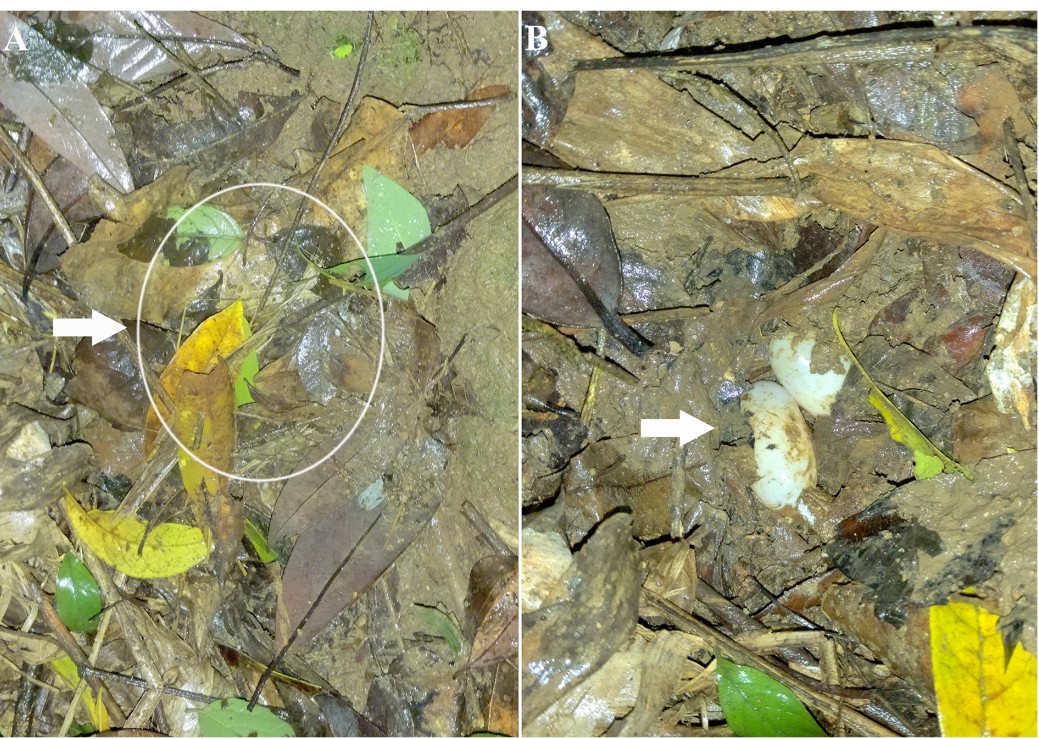

**Figure 3 Well-camouflaged nest and eggs of *S. bealei*.** (A) the nest (denoted by a circle and arrow) covered by leaves and soil; (B) two eggs inside the nest indicated by the arrow. Photo credit: Qingru Hu.

and a positive, linear correlation found between mean egg width and female carapace height (Pearson correlation coefficient = 0.847, $R^2 = 0.718$, $P < 0.05$). The mean RCM was 9.47 ± 1.01% (5.14–12.21%, $n = 6$) and the mean REM 4.60 ± 0.44% (2.98–6.11%, $n = 6$).

## Incubation and hatchling characteristics

Among the 13 eggs inside six nests, 10 were developed (76.9%, 33.3–100% per nest) and seven successfully hatched (70%, 0–100% per nest). The incubation period was 85–108 days (mean = 94.7 ± 2.5 days; $n = 7$) and the temperature range during incubation was 17.51–29.64 °C (mean = 25.08 ± 0.13 °C; $n = 7$).

The seven hatchlings had a mean weight of 9.7 ± 0.5 g, carapace length of 40.1 ± 0.5 mm, carapace width of 33.3 ± 1.4 mm, carapace height of 17.4 ± 0.5 mm, plastron length of 31.6 ± 0.5 mm, and plastron width of 25.4 ± 0.7 mm. Hatchlings had soft, reddish-brown carapaces with curled margins that gradually flattened after several days, orange plastron and skin, and bright yellow stripes on their neck. Four bright-green eye-spots were obvious on the head, with the two spots on the same side linked together. A black dot could be seen in every eye-spot (Fig. 4).

Two hatchings from the same clutch died two days after hatching, probably due to an ant infestation, as we found many ants on their body when the nest was opened. The remaining five hatchlings were kept in the laboratory until their yolk sac was absorbed, after which we released them to the stream in two days.

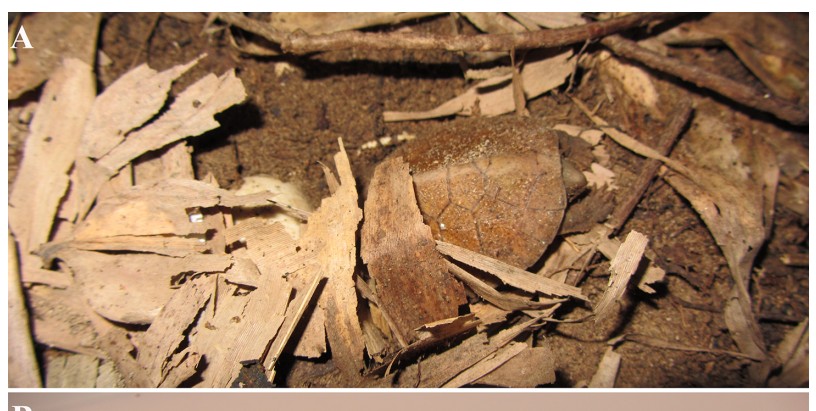

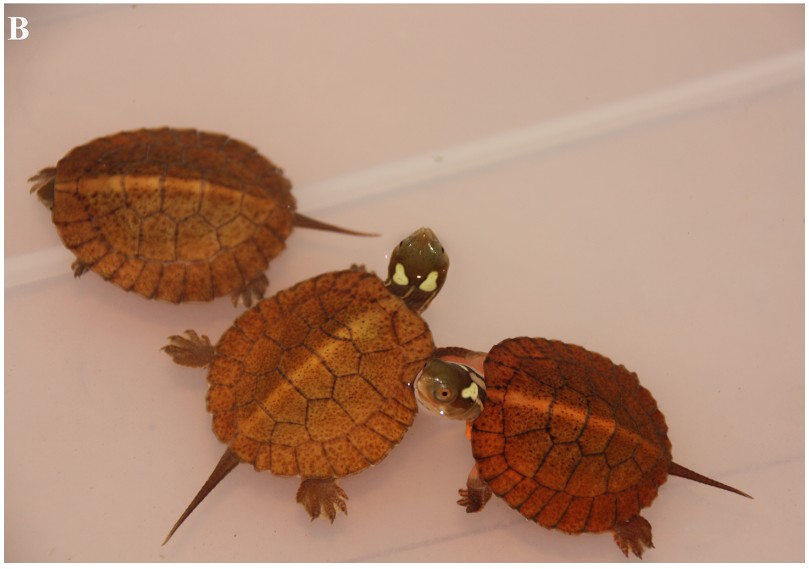

**Figure 4 Hatchlings of *S. bealei*.** (A) One hatchling emerging from the nest; (B) one-week old hatchlings. Photo credit: Qingru Hu (A), Liu Lin (B). 

## DISCUSSION

The process of selecting a nesting site is important to females because they are more vulnerable to terrestrial predators at this time (*Spencer, 2002*). Therefore, many turtle species spend less than three hours out of water during the nesting process (*Doody et al., 2009*; *Booth, 2010*). However, we found *S. bealei* would spend 4–10 h out of water when nesting. Perhaps the presence of dense forest and shrubs offers protection from predators. Additionally, the small body size and cryptic coloration (dark-brown carapace) of *S. bealei* may contribute to their safety.

Turtle body size has been shown to influence reproductive potential in female turtles (*Valenzuela, 2001*), as the area of the pelvic girdle is correlated with female size and may constrain the size of eggs an individual can oviposit (*Bowden et al., 2004*). Though considered a small-sized turtle species, *S. bealei* produces larger eggs (mean length 45.5 mm, width 23.2 mm, weight 14.8 g) than some larger freshwater species, such as *T. s. elegans* (egg length 35.4 mm, width 22.1 mm, weight 10.36 g, *Tucker & Janzen, 1998*). Continuing the comparison with *T. s. elegans*, *S. bealei* produces smaller clutches

(average 2.2 eggs) of larger eggs, while *T. s. elegans* produces larger clutches (average 12.5 eggs) of smaller eggs. Consequently, the RCM of *S. bealei* (5.14–12.21%) was similar to other freshwater turtle species: *C. mouhotii* (6.9–14.6%, *Wang et al., 2011*), *C. flavomarginata* (2.3–9.2%, *Chen & Lue, 1999*) and *T. s. scripta* (3–17%, *Ernst & Lovich, 2009*). Therefore, the total volume for eggs is relatively similar, but different species take different approaches to eggs—many small or few big. No correlations were found in our study between female body size and clutch size or egg size, however this may be due to small sample size. As we did not continue tracking turtles after nesting, we do not know whether females nested a second time during the breeding season. However, in captivity, females usually produced only one clutch per year (*Gong et al., 2017b*), implying that *S. bealei* has a low intrinsic rate of population increase.

The incubation period of *S. bealei* (94.7 ± 2.5 days at mean 25.1 °C) was relatively long when compared to *T. s. elegans* (62.25 days at mean 27.4 °C, *Yang, 2014*) and Magdalena River Turtle (*Podocnemis lewyana*) (59.1 days at mean 32.8 °C, *Correa-H et al., 2010*). This is likely due to the nest conditions—*S. bealei* nests in cooler, shaded forests, while the other two species nest in open areas with higher nest temperatures.

Both development and hatching rate of eggs (76.9% and 70%, respectively) in our study were higher than *S. bealei* in captivity (29.6% and 62.5%, respectively; *Gong et al., 2017b*). The natural habitat with lower anthropogenic disturbance may contribute the higher success of reproduction in field. Ant predation likely contributed to the death of two hatchling turtles. Ant predation is commonly reported in many other turtle species (*Parris, Lamont & Carthy, 2002*; *Ferreira Júnior et al., 2011*; *Buhlmann & Offman, 2001*; *Correa-H et al., 2010*; *Yang, 2014*; *Erickson & Baccaro, 2016*). In some cases, invertebrates including ants, flies, and beetles could infest more than 50% of nests (*Baran et al., 2001*). No evidence of vertebrate nest-predators was found in our study, probably because of the nest protection we constructed.

Successful conservation management of the endangered *S. bealei* will likely involve a combination of in-situ and ex-situ approaches. We believe the information on the reproductive ecology from our study (habitat use, nesting, breeding behavior) will help guide habitat protection and captive breeding of this rare turtle species endemic to China.

## ACKNOWLEDGEMENTS

We would like to thank Wensi Wu, Jianqing Ye, Weijiang Wu and Jianfei Ye for the logistical support and Chenwu Shen for his great contribution in field work.

### Funding

This study was supported by the National Nature Science Foundation of China (31772486), Hainan College Scientific Research Project (Hnky 2015–26) and the Croucher Foundation Chinese Visitorship. The funders had no role in study design, data collection and analysis, decision to publish, or preparation of the manuscript.

## Grant Disclosures

The following grant information was disclosed by the authors:
National Nature Science Foundation of China: 31772486.
Hainan College Scientific Research Project: Hnky 2015–26.
Croucher Foundation Chinese Visitorship.

## Competing Interests

The authors declare that they have no competing interests.

## Author Contributions

- Liu Lin conceived and designed the experiments, performed the experiments, analyzed the data, contributed reagents/materials/analysis tools, prepared figures and/or tables, authored or reviewed drafts of the paper, approved the final draft.
- Qingru Hu conceived and designed the experiments, performed the experiments, analyzed the data, contributed reagents/materials/analysis tools, prepared figures and/or tables.
- Jonathan J. Fong analyzed the data, authored or reviewed drafts of the paper, approved the final draft.
- Jiangbo Yang conceived and designed the experiments, performed the experiments, contributed reagents/materials/analysis tools.
- Zhongdong Chen conceived and designed the experiments, performed the experiments, contributed reagents/materials/analysis tools.
- Feiyu Zhou conceived and designed the experiments, performed the experiments, contributed reagents/materials/analysis tools.
- Jichao Wang contributed reagents/materials/analysis tools.
- Fanrong Xiao contributed reagents/materials/analysis tools.
- Haitao Shi conceived and designed the experiments, performed the experiments, analyzed the data, contributed reagents/materials/analysis tools, authored or reviewed drafts of the paper, approved the final draft.

## Animal Ethics

The following information was supplied relating to ethical approvals (i.e., approving body and any reference numbers):

This study was approved by the Animal Research Ethics Committee of Hainan Provincial Education Center for Ecology and Environment, Hainan Normal University (No. HNECEE-2014-003).

## Field Study Permissions

The following information was supplied relating to field study approvals (i.e., approving body and any reference numbers):

The field study was approved by Administration and Services Centre of Nanjing County, Fujian Province (No. 20141203NJ0173).

## Data Availability

The raw data and measurements are available in the Supplemental Files.

## Supplemental Information

Supplemental information for this article can be found online at http://dx.doi.org/
10.7717/peerj.4997#supplemental-information.

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
