# Peer review of "Reproductive ecology of the endangered Beal’s-eyed turtle, Sacalia bealei"

_PeerJ, doi:10.7717/peerj.4997_

## Round 0.1 · original submission · Major Revisions

Please consider carefully all of the comments of the reviewers and provide a point-by-point indication of how you have responded to each. Please pay particular attention to Reviewer 1's recommendation to take out any extraneous discussion that has nothing to do with turtle nesting, incubation or hatchlings as well as this reviewer's comment that you cannot conclude that the turtles only next once per year based on your observations. I would also encourage you to add an appropriate figure as suggested by Reviewer 2.

Reviewer 1 ·

Basic reporting

The English used is clear, I made many comments on ms for a better English style. There is no problem with the text, except they make several untrue statements, I have cut out much of the text whcih talks about conservation issues and poaching, this article is about reproduction, and does not need to have and discussion about humans trafficking this species.
They do not describe the type of transmitters used. They should have a photo of their string trailing device. I do not know why they mentioned radio telemetry if they did not report the data. Also they did not report if they followed the female turtles after they returned to the river. These turtles could have had the opportunity to nest several more times during the season. No mention was made if they followed the turtles until hibernation or not or if they exrayed them again or not. If they had used infrasound they could have detected enlarged follicles as well as oviducal eggs, which would suggest future clutches of the same year. Many format errors in Lit Cit.

Experimental design

There was no mention of turtle activity after they returned to the water, the turtles should have been followed with radio telemetry for at least 2 months and captured and palped, placing a finger into the inguinal cavity will allow one to detect the presence of shelled eggs in turtles, even species such as Kinosternon acutum, a much smaller species where nesting females have a mean crapace length of 97 mm and mass of 117 g, they can be palped, and they froduce 2-4 clutches of 1-3 eggs during a season.

Validity of the findings

The nest descriptions, number of eggs per clutch, incubation temperatures, haching and female sizes are all important findings.

But, as i said above, since they did not follow the females after they nested they can not validate that they only nest once a year

Also they do not know if the eggs that did not develop were fertile or not, they did not open the eggs and inspect the membranes for dead sperm.

But fecundity does not seem to be a problem in this study

Additional comments

I think you did a wonderful job writing this article in English. You should report the size of the females that nested in the results in the text, the measurements of the hatchlings should be in the text and the temperature data as well delete all of the tables, and it would be informative to have a photograph of a turtle with the trailing device attached. If you have data with radio telemetry after the turtles nested you should report on it. You should also mention when they entered hibernation.
I think your data are worthy of publication, heed my remarks and rewrite it. The allusion to seaturtles and larger freshwater turtles making flask shaped nests is not relevant, very small freshwater turtles also make flask shaped nests, I do not think bringing a discussion of flask shaped nests has anything to do with your study. You do not talk about sex ratios in nature, so do not mention it. This article is about nesting and incubation in nature. Leave all of the material about the pet trade out of the article.

Annotated reviews are not available for download in order to protect the identity of reviewers who chose to remain anonymous.

·

Basic reporting

The basic reporting of the paper is fine, except for the noticeable lack of figures. Since PeerJ publishes free color this is an easy way to greatly enhance the reporting of this paper.

Line 47 (and elsewhere) the authors discuss spool and line tracking, they should show images of what this is. How it looks in the field.

Lines 57 and 58 describe the vegetation, the authors should supplement the short written description with images. Show the stream and and nesting sites.

Line 138 show camoflauged nest and fishing line

Lines 160-164 they give description of hatchlings that is begging for photographs.

Lines 175 and elsewhere they discuss the nest shape, they should show pictures.

Optional images would be of the adult species.

Experimental design

I think the experimental design is fine. Maybe there are other approaches, I don't know, but they added many new data and inisghts about this poorly know species.

Validity of the findings

I think their conclusions are well stated and connect to the literature.

Additional comments

This paper definitely needs some photographs.

---

## Round 0.2 · accepted · Accept

Your revised manuscript has successfully address the comments and suggestions of both reviewers. In particular, the addition of several pictures is very informative.

#